# NXP031 prevents dopaminergic neuronal loss and oxidative damage in the AAV-WT-α-synuclein mouse model of Parkinson's disease

Min Kyung Song[ORCID][1,2], Levi Adams[1,2], Joo Hee Lee[3], Yoon-Seong Kim[ORCID][1,2,4]*

1 Robert Wood Johnson Medical School Institute for Neurological Therapeutics and Department of Neurology, Rutgers Biomedical and Health Sciences, Piscataway, NJ, United States of America, 2 Burnett School of Biomedical Sciences, UCF College of Medicine, University of Central Florida, Orlando, FL, United States of America, 3 College of Nursing Science, Kyung Hee University, Seoul, Republic of Korea, 4 Nexmos Co Ltd, Yongin-Si, Gyeonggi-Do, Republic of Korea

* yk525@rwjms.rutgers.edu

**Data Availability Statement:** All relevant data are within the paper and its Supporting Information files.

## Abstract

Parkinson's disease (PD) is a neurodegenerative disease characterized by inclusions of aggregated α-synuclein (α-Syn). Oxidative stress plays a critical role in nigrostriatal degeneration and is responsible for α-Syn aggregation in PD. Vitamin C or ascorbic acid acts as an effective antioxidant to prevent free radical damage. However, vitamin C is easily oxidized and often loses its physiological activity, limiting its therapeutic potential. The objective of this study was to evaluate whether NXP031, a new compound we developed consisting of Aptamin C and Vitamin C, is neuroprotective against α-synucleinopathy. To model α-Syn induced PD, we stereotactically injected AAV particles overexpressing human α-Syn into the substantia nigra (SN) of mice. One week after AAV injection, NXP031 was administered via oral gavage every day for eight weeks. We found that oral administration of NXP031 ameliorated motor deficits measured by the rotarod test and prevented the loss of nigral dopaminergic neurons caused by WT-α-Syn overexpression in the SN. Also, NXP031 blocked the propagation of aggregated α-Syn into the hippocampus by alleviating oxidative stress. These results indicate that NXP031 can be a potential therapeutic for PD.

## Introduction

Parkinson's disease (PD) is a common neurodegenerative disease. The neuropathological hallmarks of PD consist of the progressive loss of dopaminergic neurons in the substantia nigra (SN) and inclusions of aggregated α-synuclein (α-Syn), called Lewy bodies and Lewy neurites, in neurons [1, 2]. The Braak hypothesis postulates that α-Syn pathology can spread from the gut via the vagus nerve or anterior olfactory nucleus to the brain as the disease progresses [3]. α-Syn aggregates spread widely to the various interconnected brain regions by cell-to-cell propagation [4].

**Funding:** This work was fully supported by the Nexmos Co., Ltd., Republic of Korea. The funders had no role in study design, data collection and analysis, decision to publish, or preparation of the manuscript.

**Competing interests:** Y.S.K. is a cofounder and holds shares of equity of the Nexmos Co., Ltd. The current study was funded by the Nexmos Co., Ltd. Authors confirm that this does not alter our adherence to PLOS ONE policies on sharing data and materials.

Oxidative stress occurs by an imbalance between reactive oxygen species production and antioxidants, resulting in excessive accumulation of ROS [5]. Increased ROS generates lipid peroxidation, protein modifications, and DNA damage, leading to mitochondrial dysfunction, autophagy deregulation, oxidative DNA injury, and neuroinflammation [6]. Although the exact pathogenesis of PD has not been elucidated, increasing evidence suggests that oxidative stress caused by excessive accumulation of ROS is a critical contributor to neurodegeneration of dopaminergic neurons. Also, ROS has been shown to increase pathological aggregation of α-Syn and neurodegeneration in disorders with Lewy bodies [7–9]. Therefore, antioxidant compounds that can reduce ROS levels and mitigate aggregation are attractive options for potential therapeutic development for PD.

Ascorbic acid (vitamin C) is a natural water-soluble vitamin and plays an essential role as a powerful antioxidant against free radicals [10]. Disappointingly, many studies have reported that vitamin C intake from diet and supplements had no effect on mitigating the risk of PD [11–13]. However, recently the Swedish National March Cohort long-term longitudinal study reported the positive effects of dietary vitamin C intake on PD risk. Participants in the highest tertile of dietary vitamin C showed reduced PD risk compared with those in the lowest tertile [14]. A recent study to confirm the effect of vitamin C in the MTPT-induced PD model demonstrated that vitamin C reduced dopaminergic neuronal loss by alleviating neuroinflammation makers such as microglial responses and astrocyte activation [15]. Additionally, vitamin C concentrations remain high in the central nervous system [16–18], and vitamin C has shown neuroprotective effects in vitro [19, 20]. Although there is still debate about the link between vitamin C and PD, it is clear that vitamin C is one of the excellent potential candidates for treating PD. Despite this, there are significant hurdles in developing Vitamin C-based therapeutic agents for neurodegenerative diseases. Vitamin C is rapidly oxidized and loses its antioxidant activity in the body, and it does not freely cross the blood-brain barrier.

We recently developed a new compound, NXP031, composed of Aptamin C and vitamin C, which may overcome the shortcomings of vitamin C. Aptamin C is a single-stranded DNA aptamer that maintains a stable tertiary structure and binds explicitly to vitamin C, delaying its oxidation. We recently demonstrated that NXP031 prevented nigrostriatal degeneration in a 1-methyl-4-phenyl-1,2,3,6-tetrahydropyridine (MPTP)-induced PD model, suggesting a potential therapeutic intervention for PD [21]. Although this result was encouraging, MPTP-based models do not fully recapitulate α-synucleinopathy seen in PD. Previous work clearly show that AAV-mediated overexpression of either wild-type (WT) α-Syn or PD-related mutants (A30P or A53T α-Syn) in the SN leads to a progressive loss of nigrostriatal dopaminergic neurons and replicates many PD-like pathological features [22–26]. Here, we investigated the neuroprotective effects of NXP031 in this α-Syn overexpression model and confirmed the broad applicability of NXP031's therapeutic potential.

## Materials and methods

### Animals

24 male C57BL/6J mice (7 weeks, 18–20 g) were purchased from The Jackson Laboratory (Bar Harbor, ME, USA). All animals were allowed to acclimate to the new environment for a week and were maintained in an environment with a temperature of $22 \pm 2°C$, the humidity of $50 \pm 10\%$ under a 12-h light-dark cycle. Water and food were freely available. Mice were randomly divided into 4 groups (n = 6–8 per group): AAV-Empty + saline group (AAV-GFP group), AAV-Empty + NXP031 group (AAV-GFP + NXP031 group), AAV-WT-α-SYN + saline group (AAV-WT-α-Syn group), and AAV-WT-α-SYN + NXP031 group (AAV-WT-α-Syn + NXP031 group). All animal experiments were approved by the Institutional Animal

Care and Use Committee of the University of Central Florida (IACUC protocol # PROTO201900005).

## NXP031 preparation

Development and characterization of Aptamin C has been previously described [27]. The purified DNA aptamer was obtained from Integrated DNA Technologies (IDT, IA, USA). DNA aptamer was dissolved in folding buffer, 1 mM $MgCl_2$ in 0.01 M PBS, heated in boiling water at 90–95°C for 5 min, and cooled slowly at room temperature (RT) to fold into a tertiary structure. 1mg/ml Aptamin C stock was prepared and dilution with saline to adjust the concentration to 4mg/kg for oral gavage. L-ascorbic acid (ThermoFisher Scientific, MA, USA) was freshly mixed with a DNA aptamer in a ratio of 1:50 (w/w) before oral gavage.

## Experimental designs

Recombinant AAV vectors were produced by modification of pAAV-IRES-hrGFP (Agilent) to express wild-type human α-Syn under control of the CMV promoter. Viral particles (AAV2) were produced at the University of Iowa Viral Vector Core according to their standard operating procedures (https://medicine.uiowa.edu/vectorcore/). For viral injection, the mice were deeply anesthetized with 3% isoflurane and placed in a stereotactic instrument (Stoelting, IL, USA). After making a midline incision of the scalp and a burr hole in the skull over the appropriate injection site for the SN, all mice were stereotactically injected unilaterally into the right SN with a microinjector at a rate of 0.25 µl/min with either 2 µl of GFP-only AAV or 2 µl of WT-α-Syn AAV at a concentration of $2 \times 10^{12}$ viral genomes/mL. The needle was left in place for an additional 8 min before it was slowly withdrawn. Stereotaxic coordinates for the SN were as follows: anteroposterior (AP) -3.0 mm, mediolateral (ML) -1.4 mm, and dorsoventral (DV) -4.4 mm from the skull surface. 1 week following AAV injection, NXP031 (Vitamin C/Aptamin C 200 mg/4 mg/kg, dosage determined in previous studies [21]) was administered to the AAV-GFP + NXP031 and the AAV-WT-α-Syn + NXP031 groups via oral gavage every day for 8 weeks. The AAV-GFP and the AAV-WT-α-Syn groups were orally administered with saline. 8 weeks after NXP031 or saline treatment, rotarod test was carried out. All mice were deeply anesthetized with 3% isofluorane and were sacrificed by transcardial perfusion with saline followed by 4% PFA (Fig 1A).

## Rotarod test

The rotarod test is widely used to assess the balance and motor coordination of rodents, as previously described [21]. A rotarod machine (Ugo Basile, Coerio, Italy) was used to record the

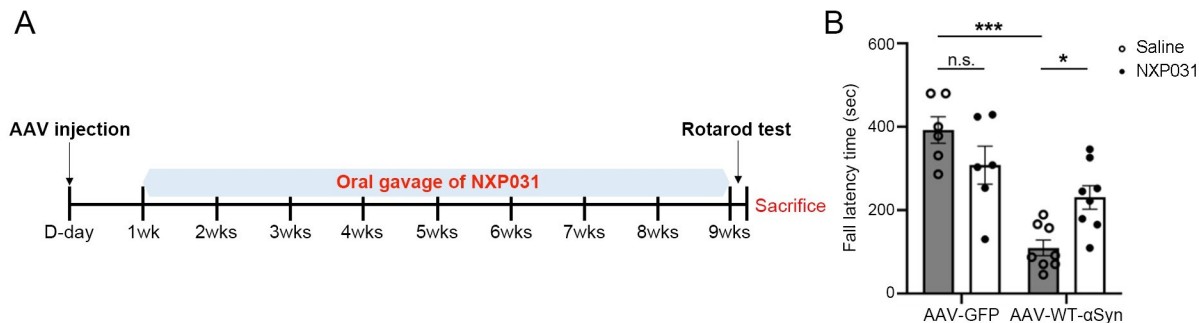

**Fig 1.** (A) Experimental paradigm. (B) Effect of NXP031 on the rotarod test at 9 weeks after delivery of GFP-only or α-Syn AAV in the SN. Latency to fall time (sec) in the rotarod test. Data are presented as the mean ± S.E.M. (Two-way ANOVA followed by Tukey's test, $^*p < 0.05$, $^{***}p < 0.001$ compared to selected group, n = 6–8 mice per group).

latency time to fall off the rotating rod. All mice were trained on the rotarod for 2 consecutive days at a fixed speed of 10 rpm for 60 s. Mice were received 3 training trials per day with a 1 h inter-trial interval. The next day, mice were tested at an automatically accelerated speed from 0 to 30 rpm. When the mice fell off the rod, the latency time to fall was recorded, up to a maximum of 480 s.

## Immunohistochemistry and immunofluorescence

The brain samples were cut at a thickness of 30 μm in the coronal plane on a freezing microtome (CM3050S; Leica, Nussloch, Germany). The brain tissues of the SN and hippocampus were selected, and the free-floating method was used for immunohistochemistry as previously described [28]. First, selected tissues were incubated with 0.3% hydrogen peroxide in 0.01 M PBS for 15 min at RT to quench endogenous peroxidase activity. Next, the tissues were blocked with 1% bovine serum albumin (BSA) in 0.01 M PBS for 1 h to reduce non-specific staining and then incubated with anti-tyrosine hydroxylase (TH, 1:500; sc-25269, Santa Cruz, CA, USA), anti-pS129 (1:200; ab51253, Abcam, MA, USA), anti-4-HNE (1:200; HNE11-S, Alpha Diagnostic International Inc, TX, USA), and anti-phosphorylation of histone H2A.X (γH2A.X) (1:500; 9718S, Cell signaling, MA, USA) diluted in 0.3% Triton X-100 and 0.5% BSA in 0.01 M PBS for overnight at 4˚C. To visualize the primary antibody, anti-mouse or anti-rabbit biotinylated secondary antibody (1:500; Vector Laboratories, CA, USA) was applied for 2 h at RT, followed by avidin–biotin complex solution (Vector Elite ABC kit; Vector Laboratories, CA, USA) for 1 h at RT. Finally, the sections were developed with a 3.3′-diaminobenzidine tetrahydrochloride (DAB kit; Vector Laboratories, CA, USA). The stained samples were mounted onto slides, coverslipped, and imaged under an optical microscope (Leica DMi8; Leica, Bensheim, Germany). To quantify pS129-α-Syn, 4-HNE, and pH2A.X staining in the hippocampus, pS129-α-Syn, 4-HNE, and γH2A.X positive cells were counted in 2 sections per mouse. Cell counting was performed blind to the identity of groups by two researchers who did not participate in this experiment. The two researchers' average values were taken as the representative values.

Fluorescence immunostaining was performed in the same procedure as above for immunohistochemistry. As secondary antibodies, the Alexa 488 conjugated anti-mouse (1:500; Invitrogen, CA, USA) and Alexa 647 conjugated anti-rabbit (1:500; Invitrogen, CA, USA) were used. Tissue sections were then stained with Hoechst (Tocris Bioscience, MO, USA) and coverslipped for imaging using a Dragonfly 2000 confocal microscope.

## Unbiased stereological counting

For unbiased stereological counting, we estimated the number of TH-positive neurons in the SN using Leica DM4B motorized microscope equipped with StereoInvestigator software (MicroBrightField Bioscience, Williston, VT, USA). The total number of TH-positive neurons was estimated according to the optical fractionator workflow probe. Every sixth section (30 μm thickness), from the anterior to the posterior midbrain, was analyzed. The ipsilateral SN was first outlined at low magnification (5x). The counting frame size was set 60 x 60 μm, the grid frame size was set 120 x 120 μm. Top and bottom guard zones were applied to each site, and an optical dissector height of 25 μm was used. Counts for TH-positive neurons were performed at high magnification (20x). An acceptable cell estimation had a coefficient of error (CE) using the Gunderson method (m = 1) less than 0.1.

## Nissl staining

The brain tissues of SN were stained with FD cresyl violet solution (FD Neurotechnologies, Inc., MD, USA) and rinsed briefly in distilled water, then differentiated in 95% ethanol

containing 0.1% glacial acetic acid for 1 min. The tissues were dehydrated in 100% ethanol for 2 min and placed in xylene, then coverslipped.

## Statistical analysis

Statistical analyses were performed using GraphPad Prism 9.3.1 (GraphPad Software Inc., CA, USA). Two-way analysis of variance (ANOVA) was used in comparing means of all the groups, followed by Tukey's *post-hoc* tests for multiple comparisons. All data were presented as means ± S.E.M. The criterion for significance was set at $p < 0.05$.

## Results

### NXP031 attenuates motor deficits induced by AAV-mediated overexpression of human α-Syn in the SN

To model dopaminergic degeneration caused by α-synucleinopathy, AAV particles containing human α-Syn or GFP-only control were stereotactically injected unilaterally into the SN of mice. We performed TH immunostaining to check whether there was dopaminergic neuronal loss in the SN 1 week after the AAV-WT-α-Syn injection, and found no dopaminergic neuronal loss (S1 Fig). 1 week after AAV injection, NXP031 or saline was administered via oral gavage every day for 8 weeks. 9 weeks after delivery of AAV, we evaluated motor function using the rotarod test (Fig 1B). A two-way ANOVA revealed that there was a statistically significant interaction between the effects of AAV vector type (GFP-only or WT-α-Syn) and treatment (saline or NXP031) ($F_{1, 24} = 11.00$, $p = 0.003$). Mice in the AAV-WT-α-Syn group showed a shorter latency to fall than the AAV-GFP injected mice ($p < 0.001$), indicating that α-Syn overexpression led to motor deficits. However, the AAV-WT-α-Syn + NXP031 group exhibited a significant increase in latency to fall compared with the AAV-WT-α-Syn group ($p = 0.031$). Our finding demonstrates that NXP031 prevented motor dysfunction caused by α-Syn overexpression in the SN.

### NXP031 blocks the loss of dopaminergic neurons against AAV-mediated overexpression of human α-Syn in the SN

At 9 weeks post-injection, we checked whether our AAV-GFP titer was neurotoxicity in the SN. The number of TH-positive neurons between the contralateral and ipsilateral sides was compared, showing no significant difference with slightly less number in the ipsilateral SN (S2 Fig) (t = 1.787, $p = 0.104$). To assess the protective effect of NXP031 on dopaminergic neurons/fibers in the striatum and SN, we performed TH immunohistochemistry (Fig 2A–2D). We observed significant differences in the optical density of TH-stained dopaminergic fiber in the striatum (a main effect of vector group $F_{1, 24} = 50.00$, $p < 0.001$; a main effect of treatment $F_{1, 24} = 4.561$, $p = 0.043$; without a significant interaction) and in the survival of nigral dopaminergic neurons in the SN (AAV vector type x treatment interaction $F_{1, 24} = 21.81$, $p < 0.001$). We observed drastic loss of dopaminergic neurons in the AAV-WT-α-Syn group, showing a 55–60% loss of TH-positive neurons in the SN compared to the AAV-GFP injected mice ($p < 0.001$). Interestingly, oral gavage of NXP031 for 8 weeks showed a significant protective effect against dopaminergic neuronal degeneration, preserving up to 65% of TH-positive neurons in the AAV-WT-α-Syn + NXP031 group compared to the AAV-WT-α-Syn group ($p < 0.001$) (Fig 2E). To confirm that the decreased TH immunoreactivity was caused by neuronal loss and not by transient decreases in TH expression changes, we performed Nissl staining (Fig 2F and 2G). We observed a similar reduction in the number of Nissl-positive cells in the SN at 9 weeks after α-Syn overexpression. Together with decreased TH-positive staining,

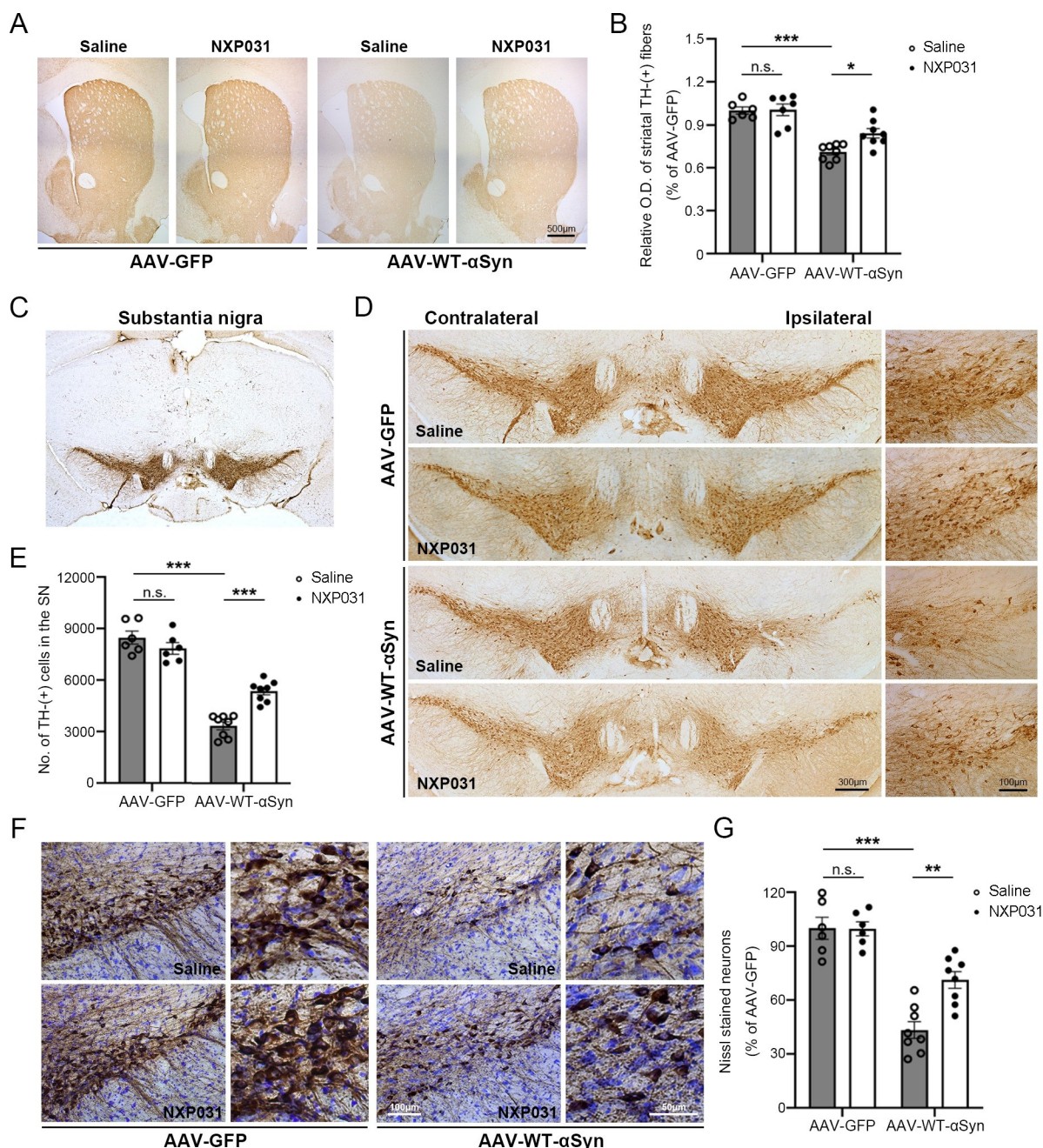

**Fig 2. Effect of NXP031 on TH-immunoreactivity in the striatum and SN at 9 weeks after delivery of GFP-only or α-Syn AAV.** (A) Representative micrographs of TH immunostaining in the striatum. (B) The relative optical density of TH-stained dopaminergic fiber in the striatum (C, D) Representative micrographs of TH immunostaining in the SN. (E) The number of TH-positive neurons in the SN. (F) Representative micrographs of double staining with TH antibody and Nissl violet in the SN. (G) % of Nissl-positive neurons compared to the AAV-GFP group. Data are presented as the mean ± S.E.M. (Two-way ANOVA followed by Tukey's test, $^*p < 0.05$, $^{**}p < 0.01$, $^{***}p < 0.001$ compared to selected group, n = 6–8 mice per group).

this result indicates dopaminergic neuronal degeneration (AAV vector type x treatment interaction $F_{1, 24} = 8.187$, $p = 0.009$). The result implies that NXP031 efficiently blocked dopaminergic neuronal death caused by α-Syn overexpression in the SN.

## NXP031 prevents accumulation and propagation of phosphorylated α-Syn in the brain

Phosphorylation of α-Syn at the serine 129 residue (pS129-α-Syn) is known to be associated with the pathological changes of PD and enhances fibril formation and insoluble aggregation of α-Syn [29, 30]. We performed double immunostaining of TH and pS129-α-Syn, demonstrating that pS129-α-Syn staining is mostly co-stained with TH-positive neurons (Fig 3A). No pS129-α-Syn staining was observed in the AAV-GFP group, while the AAV-WT-α-Syn injected mice displayed pronounced staining of phosphorylated α-Syn in the SN (Fig 3B). However, contrary to our expectations, we observed that pS129-α-Syn levels were reduced in the areas with the highest dopaminergic neuronal loss. As previously reported, severe dopaminergic neuronal loss accounts for concomitant loss of cellular markers such as pS129-α-Syn [31]. However, this made it challenging to simply quantify pS129-α-Syn levels in the SN to assess the effects of NXP031 on α-Syn aggregation.

To circumvent this issue, we examined the hippocampus of our mice. Propagation of α-Syn aggregates from the SN to the hippocampus over time is well known [32–34]. Our Nissl staining in the hippocampus showed that hippocampal neuronal loss was minimal even if α-Syn pathology was observed. We found significant differences in pS129-α-Syn levels in the hippocampal dentate gyrus (DG) and cornu ammonis 3 (CA3) regions (DG: AAV vector type x treatment interaction $F_{1, 24} = 4.570$, $p = 0.043$; CA3: AAV vector type x treatment interaction $F_{1, 24} = 9.716$, $p = 0.005$) (Fig 3C). The number of pS129-α-Syn-positive neurons in the AAV-WT-α-Syn group was largely increased compared to the AAV-GFP group (DG: $p = 0.002$; CA3: $p < 0.001$). However, oral administration of NXP031 significantly ameliorated the pS129-α-Syn levels in the hippocampal regions compared to the AAV-WT-α-Syn group (DG: $p < 0.001$; CA3: $p < 0.001$) (Fig 3D and 3E), and appeared similar to GFP-only (DG: $p = 0.988$, CA3: $p = 0.285$). The result suggests that the hippocampal regions are relatively resistant to α-Syn-mediated cell death, and NXP031 can effectively prevent the propagation of α-Syn phosphorylation and aggregation into the hippocampus.

## NXP031 blocks oxidative stress and DNA double-strand breaks

4-hydroxynonenal (4-HNE) is a product of lipid peroxidation and is considered a biomarker of oxidative stress [35]. To estimate the oxidative stress induced by α-Syn overexpression, we measured the levels of 4-HNE expression in the SN and hippocampus (Fig 4A and 4B). 4-HNE immunoreactivity was similar to that of pS129-α-Syn. 4-HNE staining was not detected in the SN region with severe dopaminergic neuronal loss, so we examined the hippocampal regions. Two-way ANOVA of the number of 4-HNE positive cells in the hippocampal regions showed a significant difference among the whole experimental groups (DG: AAV vector type x treatment interaction $F_{1, 24} = 6.557$, $p = 0.017$; CA3: AAV vector type x treatment interaction $F_{1, 24} = 5.325$, $p = 0.030$). Compared with the AAV-GFP group, the AAV-WT-α-Syn group showed a dramatic increase in the number of 4-HNE positive cells in the hippocampal regions (DG: $p < 0.001$; CA3: $p < 0.001$). NXP031 treatments significantly reduced 4-HNE generation induced by the propagation of aggregated-α-Syn in the hippocampus (DG: $p = 0.003$; CA3: $p = 0.011$), with levels similar to control (DG: $p = 0.642$; CA3: $p = 0.276$) (Fig 4C and 4D).

Aside from lipid peroxidation, oxidative stress can also lead to DNA damage. γH2A.X occurs in response to DNA double-strand break formation as an indicator of DNA damage [36]. We investigated the levels of γH2A.X in the SN and hippocampus and found that its levels were well correlated with the 4-HNE staining pattern (DG: AAV vector type x treatment interaction $F_{1, 24} = 4.729$, $p = 0.040$; CA3: AAV vector type x treatment interaction $F_{1, 24} = 7.154$, $p = 0.013$) (Fig 5A and 5B). The AAV-WT-α-Syn group showed a marked increase in the

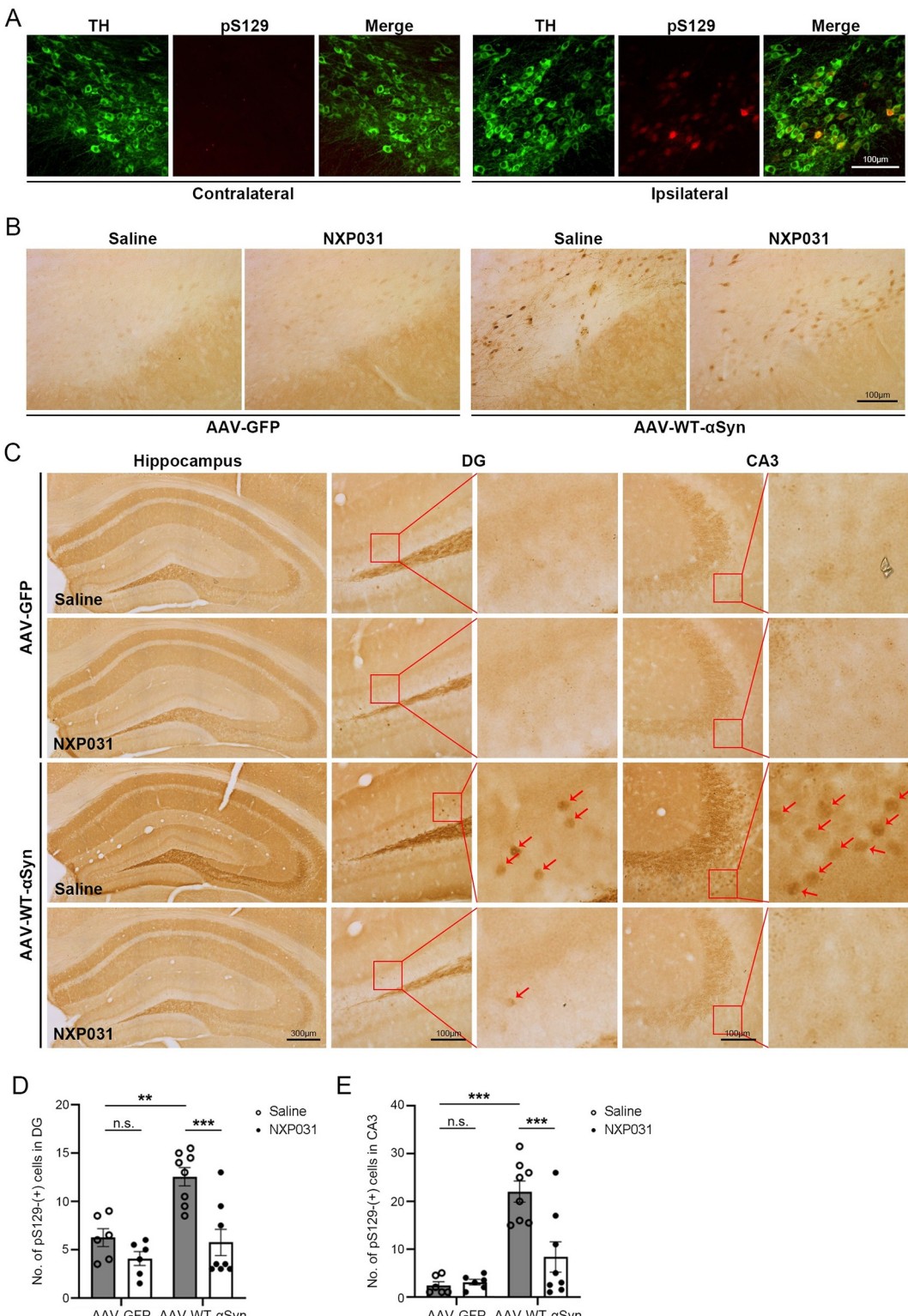

**Fig 3. Effect of NXP031 on pS129-α-Syn expression in the SN and hippocampus after delivery of GFP-only or α-Syn AAV.** (A) Representative micrographs of double immunostaining of TH and pS129-α-Syn in the SN. (B) Representative micrographs of pS129-α-Syn immunostaining in the SN. (C) Representative micrographs of pS129-α-Syn immunostaining in the hippocampus. (D) The number of pS129-α-Syn-positive cells in the hippocampal DG. (E) The number of pS129-α-Syn-positive cells in the hippocampal CA3. Data are presented as the mean ± S.E.M. (Two-way ANOVA followed by Tukey's test, $^{**}p < 0.01$, $^{***}p < 0.001$ compared to selected group, n = 6–8 mice per group).

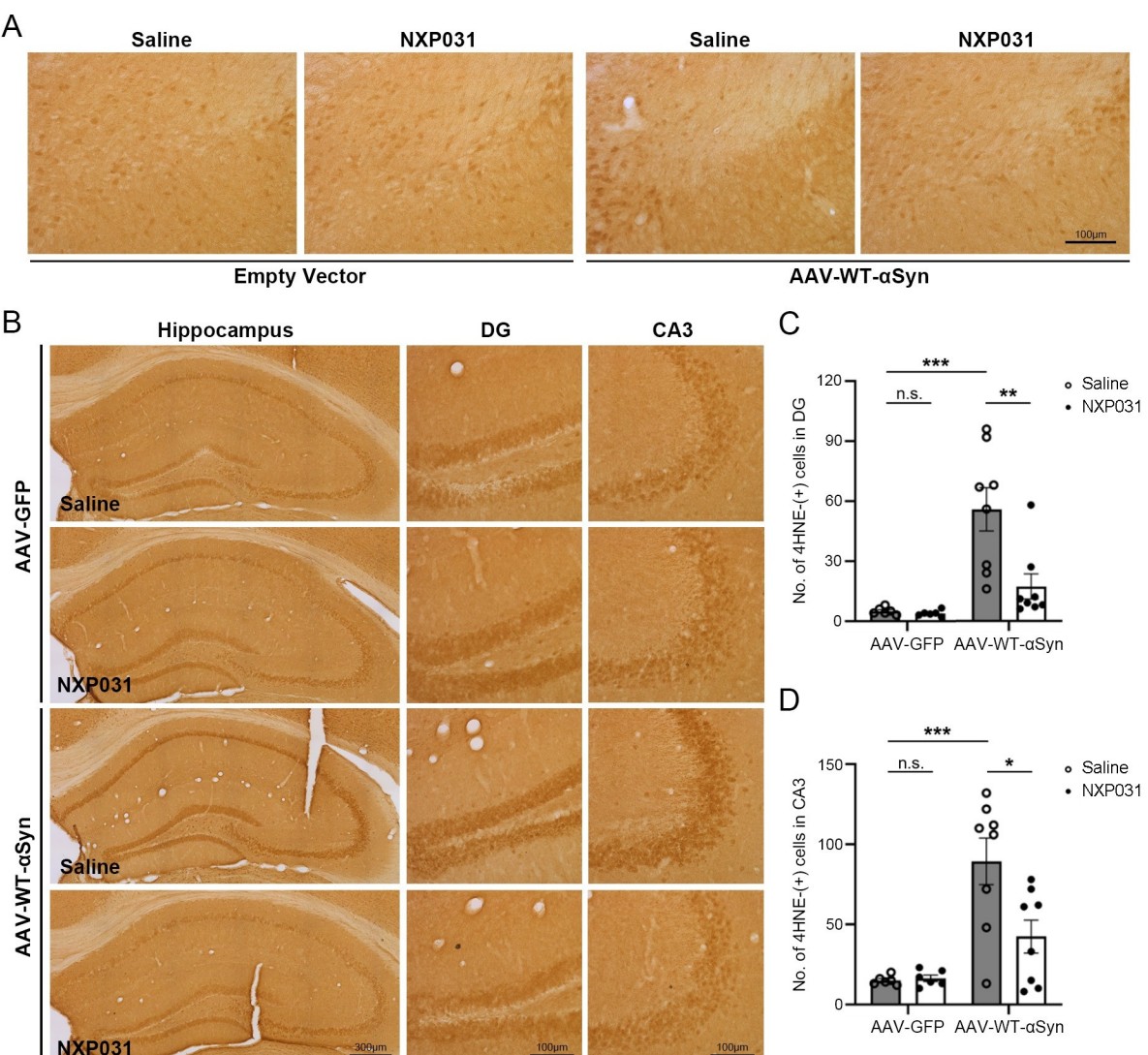

**Fig 4. Effect of NXP031 on oxidative stress in the SN and hippocampus of mice after delivery of GFP-only or α-Syn AAV.** (A) Representative micrographs of 4-HNE immunostaining in the SN. (B) Representative micrographs of 4-HNE immunostaining in the hippocampus. (C) The number of 4-HNE positive cells in the hippocampal DG. (D) The number of 4-HNE positive cells in the hippocampal CA3. Data are presented as the mean ± S.E.M. (Two-way ANOVA followed by Tukey's test, $^*p < 0.05$, $^{**}p < 0.01$, $^{***}p < 0.001$ compared to selected group, n = 6–8 mice per group).

number of γH2A.X-positive cells and stronger immunostaining in the hippocampus than the AAV-GFP group (DG: $p < 0.001$; CA3: $p < 0.001$). Administration of NXP031 inhibited DNA damage in the hippocampus, leading to a significant decrease in the number of γH2A.X-positive cells (DG: $p = 0.016$; CA3: $p = 0.003$), and was not significantly different than GFP-only (DG: $p = 0.477$; CA3: $p = 0.171$) (Fig 5C and 5D). These results suggest that NXP031 prevents oxidative stress and related DNA damage caused by α-Syn aggregate propagation.

## Discussion

α-Syn is a central protein in PD pathology. PD is characterized by the loss of dopaminergic neurons in the SN and accumulation of aggregated of α-Syn in dopaminergic neurons [1, 37].

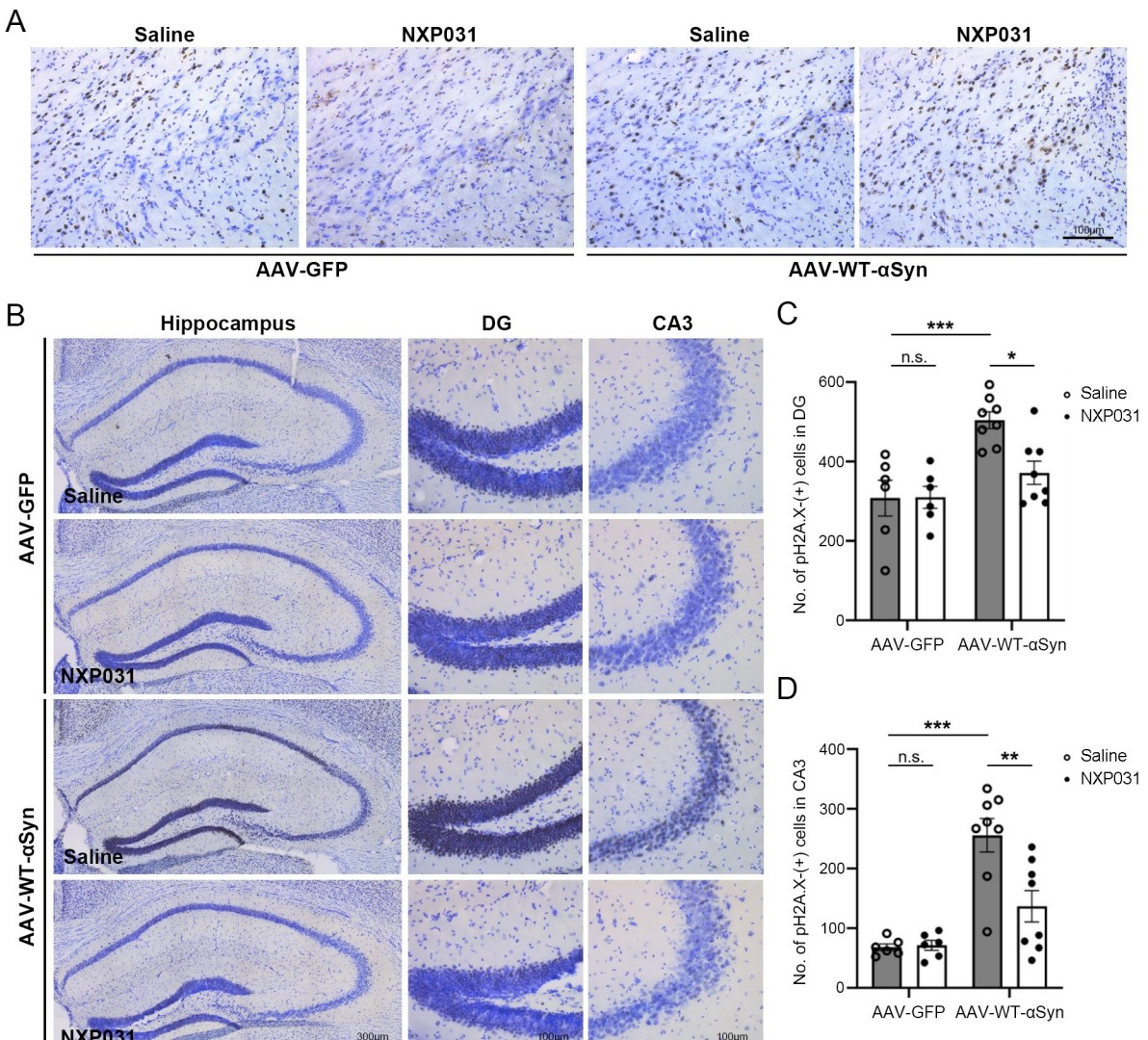

**Fig 5. Effect of NXP031 on DNA damage in the hippocampus of mice after delivery of GFP-only or α-Syn AAV.** (A) Representative micrographs of γH2A.X immunostaining in the SN. (B) Representative micrographs of γH2A.X immunostaining in the hippocampus. (C) The number of γH2A.X positive cells in the hippocampal DG. (D) The number of γH2A.X positive cells in the hippocampal CA3. Data are presented as the mean ± S.E.M. (Two-way ANOVA followed by Tukey's test, $^*p < 0.05$, $^{**}p < 0.01$, $^{***}p < 0.001$ compared to selected group, n = 6–8 mice per group).

Thus, therapeutic interventions targeting abnormal α-Syn aggregation might be promising in preventing or slowing down the degeneration process of PD. In this study, we investigated the neuroprotective effects of NXP031, a compound combining Aptamin C and vitamin C, under neurodegenerative conditions induced by AAV-mediated overexpression of human α-Syn in the SN.

At 9 weeks after the human WT-α-Syn/AAV-injection into the SN, we assessed the motor function and dopaminergic neuronal survival at the injection site. Consistent with previous studies, AAV-mediated α-Syn overexpression in the SN of mice led to a significant reduction of TH-positive neurons, leading to motor dysfunction [38–40]. To make sure dopaminergic neuronal loss, we also used Nissl staining, a observed a loss of both TH and Nissl staining indicating dopaminergic neuronal degeneration. NXP031 treatments ameliorated motor deficits

and dopaminergic neuronal death in the SN. These results support our previous study that NXP031 protects acute degeneration of nigral dopaminergic neurons in the MPTP-induced mitochondrial oxidative stress PD model [21] and expand its application to α-synucleinopathy-based PD models.

α-Syn aggregation is a biomarker for PD [2, 37]. Although dopaminergic neurons in the SN appear to be selectively vulnerable to α-Syn-induced degeneration, α-Syn pathology is not restricted to this region and propagates throughout the brain as PD progresses. In 2003, Braak and colleagues originally proposed the hypothesis that the aggregated α-Syn may start in the digestive tract and spread toward the central nervous system via the vagus nerve [3]. Eventually, the aggregated α-Syn arrives at the SN and extends to other brain areas. pS129-α-Syn is a reliable marker of α-Syn aggregates [41, 42]. We performed pS129-α-Syn immunostaining in the SN to assess the effects of NXP031 on the levels of α-Syn aggregation. AAV-mediated α-Syn overexpression resulted in pS129-α-Syn expression in the injection site. However, as dopaminergic neuronal loss in the SN increases, pS129 staining is concomitantly reduced [33]. This made it difficult to use pS129-α-Syn as an indicator of therapeutic efficacy of NXP031 for α-Syn aggregation in the SN. To circumvent this problem, we investigated other brain regions relatively resistant to α-Syn-mediated cell death. As previous studies have shown that α-Syn aggregates propagate from the midbrain to the hippocampus [32–34], we performed pS129-α-Syn staining in this region. Interestingly, the AAV-WT-α-Syn group showed significantly increased pS129-α-Syn levels compared to the AAV-GFP group even in this more distal region, which was blocked by NXP031 treatments, further highlighting the therapeutic potential of this compound.

Oxidative stress is one of the main factors contributing to dopaminergic neuronal degeneration in PD, inducing oxidative damage to proteins, lipids, and DNA [43, 44]. We then showed that NXP031 blocked oxidative stress induced by α-Syn overexpression in the SN. 4-HNE staining for toxic products of oxidative damage is a widely used biomarker for increased ROS levels, and we found that overexpression of α-Syn led to increased levels of 4-HNE staining. A previous study with immunohistochemical studies has indicated that 4-HNE-modified proteins were significantly increased in nigral melanized neurons in human post-mortem brain PD samples [45]. The A53T-α-synuclein rat model of PD has shown an increase in the 4-HNE staining signal in the SN of the AAV-A53T-a-Syn injected rat [39]. Previous studies are consistent with our results [39, 45], suggesting that oxidative stress may contribute to nigral neuronal cell death. Similar to pS129-α-Syn staining, severe dopaminergic neuronal loss caused by α-Syn overexpression made it difficult to detect 4-HNE in the SN. Alternatively, we found a significant increase in 4-HNE levels in the hippocampus caused by α-Syn aggregation's propagation into the hippocampus. We observed that NXP031 treatments prevented the propagation of aggregated α-Syn into the hippocampus, resulting in a decreased level of 4-HNE expression in the hippocampus.

DNA double-strand breaks are the most harmful DNA damage, triggering activation of phosphorylation of the histone variant H2A.X, a significant component of DNA damage response [36]. In two different synucleinopathy models of PD, DNA damage markers such as γH2A.X, 53BP1, and pATM were upregulated, demonstrating that α-Syn-mediated oxidative stress results in DNA double-strand breaks [46]. We obtained γH2A.X staining results similar to 4-HNE staining in the SN and hippocampus. While PD has strong connections with reactive oxygen species, other neurodegenerative conditions such as Alzheimer's disease and multiple amyotrophic lateral sclerosis have also been shown to have oxidative damage components. It will be worthwhile to investigate the potential of this compound on those conditions in future studies.

It is interesting to note that the oral route of administration of NXP031 exerts therapeutic effects in PD model. As increasing evidence suggests the critical contribution to the gut microbiome to PD pathogenesis, it will be interesting to investigate the effects of NXP031 on the intestinal flora and their metabolism. Combined with our previous study demonstrating that NXP031 can significantly prevent dopaminergic neuronal degeneration induced by the MPTP-induced PD model, the current study provides a strong foundation for deeper investigation into the potential benefits of this novel therapeutic strategy for PD. This promising initial result based on motor function and histological results is exciting, and in future studies we will expand our scope to include behavioral and cognitive tests. It will be interesting to see if similar protection can be seen against other PD symptoms such as cognitive decline.

In conclusion, our results demonstrate the neuroprotective promise of NXP031 in the AAV-WT-α-Syn mouse model of PD by preventing dopaminergic neuronal loss in the SN and inhibiting the propagation of α-Syn pathology further into the other brain regions through reducing oxidative stress. Thus, we suggest that NXP031 could represent a prospective therapeutic strategy for PD.

## Supporting information

**S1 Fig. Assessing dopaminergic neurons in the SN 1 weeks after delivery of AAV-WT-α-Syn.** Representative micrographs of TH immunostaining in the SN.
(TIF)

**S2 Fig. Comparison of TH-positive neurons between the contralateral and ipsilateral sides in the SN at 9 weeks after delivery of GFP-only AAV.** (A) Representative micrographs of TH immunostaining in the SN. (B) % of TH-positive neurons in the SN compared to the contralateral side. Data are presented as the mean ± S.E.M. (Student t-test, n = 6 mice).
(TIF)

## Author Contributions

**Conceptualization:** Min Kyung Song, Yoon-Seong Kim.

**Data curation:** Min Kyung Song, Levi Adams.

**Formal analysis:** Min Kyung Song.

**Investigation:** Min Kyung Song, Levi Adams, Joo Hee Lee.

**Methodology:** Min Kyung Song, Levi Adams.

**Project administration:** Yoon-Seong Kim.

**Supervision:** Yoon-Seong Kim.

**Writing – original draft:** Min Kyung Song.

**Writing – review & editing:** Levi Adams, Yoon-Seong Kim.

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
