## [Decision Letter · Decision Letter 0]

29 Mar 2022

PONE-D-22-01833NXP031 reduces dopaminergic neuronal loss and oxidative damage in the AAV-WT-α-synuclein mouse model of Parkinson’s diseaPLOS ONE

Dear Dr. Kim,

Thank you for submitting your manuscript to PLOS ONE. After careful consideration, we feel that it has merit but does not fully meet PLOS ONE’s publication criteria as it currently stands. Therefore, we invite you to submit a revised version of the manuscript that addresses the points raised during the review process.Reviewers find that your manuscript is interesting and may have therapeutic potential. However, they raised several issues that need your attention. Data from the striatum, where aSyn injected in substantia nigra is eventually transported. aSyn aggregation and  some antioxidant response markers in the substantia nigra.In addition you need to demonstrate how far the AAV-alpha synuclein spread throughout the substantia nigra. You also needed to remove a group of animals 1 week after the AAV-a-syn infusion, and prior to the start of the NXP treatment, to determine if there was any loss of TH labeled neurons in the SNpc.  There are a  number of other comments made by reviewers.

 Please submit your revised manuscript May 09 2022 11:59PM. If you will need more time than this to complete your revisions, please reply to this message or contact the journal office at plosone@plos.org. Please include the following items when submitting your revised manuscript:A rebuttal letter that responds to each point raised by the academic editor and reviewer(s). You should upload this letter as a separate file labeled 'Response to Reviewers'.A marked-up copy of your manuscript that highlights changes made to the original version. You should upload this as a separate file labeled 'Revised Manuscript with Track Changes'.An unmarked version of your revised paper without tracked changes. You should upload this as a separate file labeled 'Manuscript'.

We look forward to receiving your revised manuscript.

Kind regards,

Hemant K. Paudel

Academic Editor

PLOS ONE

Journal Requirements:

( I have read the journal's policy and the authors of this manuscript have the following competing interests [Yoon-seong Kim is a cofounder of the Nexmos Co., Ltd. Other authors have declared that no competing interests exist])  

We note that you received funding from a commercial source: [Name of Company]

Reviewers' comments:

Reviewer's Responses to Questions

**Comments to the Author**

1. Is the manuscript technically sound, and do the data support the conclusions?

Reviewer #1: Partly

Reviewer #2: Partly

2. Has the statistical analysis been performed appropriately and rigorously? 

Reviewer #1: Yes

Reviewer #2: Yes

3. Have the authors made all data underlying the findings in their manuscript fully available?

Reviewer #1: No

Reviewer #2: Yes

4. Is the manuscript presented in an intelligible fashion and written in standard English?

Reviewer #1: Yes

Reviewer #2: Yes

5. Review Comments to the Author

Reviewer #1: In the current manuscript, Kim et al. describe the effects of novel vitamin C derivative, NXP-031, on a Parkinson's disease mouse model based on AAV-alpha-synuclein injection on substantia nigra. Although the results are interesting, particularly the protective effect of NXP-031 on TH+ neuron count, there are several issues that needs to be addressed.

- My main concern is that the authors do not show any data from the striatum, where aSyn injected in substantia nigra is eventually transported? That is anyway the equally important area (or in early disease even more important) as substantia nigra in Parkinson’s disease pathology, and most of the endogenous aSyn is located in presynaptic area. I do understand hippocampus as one of the projection areas of substantia nigra where aSyn is also seen but why the authors did not study striatum and e.g. motor cortex that are more important for locomotor deficits. Moreover, as the authors use unilateral model of AAV-aSyn, a model measuring unilateral deficits e.g. cylinder test or staircase test would be probably more informative than rotarod assay, and gives usually more information on striatal damages than rotarod assay.

- Additionally, the authors do not provide any information about the aSyn aggregation itself, particularly when the pS129 aSyn did not show any reliable staining in the substantia nigra. If they consider that the mechanism-of-action for NXP031 is antioxidant mechanism, they should also show some antioxidant response markers in the substantia nigra, e.g. SOD1, catalase.

- The authors should also revise the manuscript more carefully, now there are several mistakes that make the interpretation of the results difficult: Figure 3, there are no details in the materials and methods about the pS129 aSyn analysis and that should be added. How many sections and brains were counted etc. The staining figures of pS129 aSyn also look puzzling. Why there is high amount of staining in the substantia nigra pars reticulata? I have some doubts about the antibody specificity, it appears to detect either normal aSyn as well or then some other unspecific protein. Moreover, Figure 5 should present the immunostaining of phosphorylated H2A in the hippocampus but it appears to present a Nissl staining? This needs to be corrected.

Other comments:

Introduction:

Page 3: “α-Syn aggregates spread from the gut via the vagus nerve or anterior olfactory nucleus to the brain as the disease progresses [3].” This is indeed seen in the experimental animals and certain support in clinical disease comes from Braak staging (2003) but I would still avoid such strong statements on this as aSyn may well start aggregating in brain neuronal cells without propagating species.

Page 3.” Additionally, vitamin C concentrations remain high in the central nervous system [15].” In the refence 15 (Nualart et al.) there were no neuronal cells used, and the publication is completely performed by using cell cultures and primary cells. How the authors can claim based on this reference that vitamin C concentrations remain high in the CNS?

Page 3. Several studies have shown that dietary supplements of vitamin C are not protective for PD, and authors should be careful not to draw conclusions about the beneficial effects of vitamin C on PD based on one study that has also raised some concerns (Tomoyuki Kawada; Reader Response: Dietary Antioxidants and the Risk of Parkinson Disease: The Swedish National March Cohort, 2021).

Materials and methods:

Page 4-5: Why there is no AAV-empty + NXP031 group? AAV-GFP can also be toxic for the TH+ neurons (Albert et al. J Neurosci Res 2019), was this seen in the GFP-group?

Page 5-6: How does the saline correlate with NXP031 solvent? Was NXP031 also dissolved to saline in the end?

Page 8: TH+ neuron count. More details are needed. Was stereology taken into account as the authors used 30 um sections?

Results:

Figure 4. The authors state that they didn’t detect 4-HNE staining in SNPc but in the Figure 4, there is clearly some staining visible. Why this was not analyzed or did the authors conclude that this is not specific?

Discussion:

TH loss does not always refer to loss of DAergic neurons, and this is also visible in Suppelementary Figure S1. This should be taken into account in the discussion.

Reviewer #2: March 22, 2022

Song et al report that 1 week following unilateral infusion of AAV-a-syn into the SN, daily oral administration of NXP031 for the next 8 weeks is neuroprotective in terms of blocking the alpha synuclein-induced motor dysfunction as determined using the rotorod, and blocking the alpha synuclein-induced loss of TH neurons in the SNpc. Although there was no evidence that NXP blocked the aggregation of alpha synuclein within the SNpc, since the aggregation was apparently located in non-neuronal cells, the authors quantified the aggregation within the hippocampus. This alpha synuclein-induced aggregation was blocked by prior administration of NXP within both the DG and CA3. Only within the hippocampus did NXP block the accumulation of 4-HNE, a marker of oxidative stress, and H2A.X (histone phosphorylation). Since this is a neuroprotective study, it is more appropriate to use the term, blocked, versus ameliorated. Although the aggregation extended from the SN to the hippocampus, the authors provide no behavioral data to suggest this aggregation was deleterious, which is a major limitation of this study. Also, NXP blocked the aggregation of alpha synuclein and did not reduce the aggregation. 'Reducing' implies that there was previous aggregation but this is incorrect, since this was a neuroprotective study design. In addition, since here are no prodromal biomarkers that can diagnose PD years prior to the emergence of the motor dysfunction, a neuroprotective study design is not translational to the clinic at this time. Therefore, the authors have over-interpreted their findings that NXP is a potential therapeutic for PD.

1.) Abstract: based on the comments above, the authors need to change the wording of the abstract (ie., reduced) and decrease the emphasis of the last sentence.

2.) Materials: Animals: the authors need to include an NXP only treated group that was originally infused with the AAV-empty vector group to determine if the drug has its own effects. Therefore, the 3 groups analyzed in this study is incomplete. Was the saline also administered orally? This should be included.

3.) Experimental design: The authors need to demonstrate how far the AAV-alpha synuclein spread throughout the SN. Using the GFP-only AAV, this needs to be carried out. The concern of this reviewer is that the authors needed to remove a group of animals 1 week after the AAV-a-syn infusion, and prior to the start of the NXP treatment, to determine if there was any loss of TH labeled neurons in the SNpc. This group needs to be included.

4.) Rotorod test: when did the animals undergo the 3 days of training sessions? Assume prior to the start of the AAV-infusion? Please clarify.

5.) Immunohistochemistry: although it was later explained that following the infusion of AAV-a-syn, the protein is distributed/transported to various regions, including the hippocampus, a clear experimental design flaw in this study was that there were no behavioral experiments to determine if the aggregation, increase in ROS and double stranded breaks in the hippocampus lead to a deficit in learning/memory. Considering the authors carried out rotorod testing, was there aggregation within the motor cortex? The hippocampal data, although of interest, has little to do with the neuroprotective effects in the SNpc.

For the SNpc/TH analysis, was the entire rostral-caudal extent of the SN sectioned and analyzed? This is important since it's not clear how far the AAV infusion spread along the SN and whether the infusion spread outside the SN. For the hippocampus, how many sections were analyzed?

6.) TH-positive cells counting: please provide more detail as to how the cell count was carried out. Was this done stereologically?

7.) Statistical analysis: please include the Degrees of Freedom in your ANOVA. In addition, with the needed inclusion of a 4th experimental group (VEH+NXP), the authors will need to carry out a 2-way ANOVA.

8.) Where is the Figure Legend for Figure #1? Please include.

9.) Line 191: were there rotorod motor difficulties 1 week following the infusion of AAV-a-syn but prior to the start of the NXP treatment? That would be important data to know.

10.) Line 200: similar to the comment in #9 above, how much loss of TH neurons were there 1 week after AAV-a-syn infusion but prior to the start of NXP treatment. Also please change the heading of that section to 'NXP blocks ....'.

11.) Line 206: in the S1 Figure, since these are double stained sections for TH and CV, the authors need to quantify the number of TH negative/CV positive neurons and not just show the photos. In the S1 figure, far right photographs, it appears there is loss of TH+ cells on the contralateral (left) side of the SNpc. The authors now need to also quantify the number of TH+ neurons on the contralateral side since it appears that there was possibly spread of the AAV-a-syn to the contralateral SN. In this S1 figure, the authors need to show a photo of the vehicle only control group.

12.) Line 224: the authors need to double label for p-129 and TH to determine if the p-129 is located in non-dopamine neurons.

13.) Line 230: as mentioned before, the lack of any learning/memory correlate to the increased aggregation in the DG/CA3 limits the interpretation of this finding and has little to do with the main theme of this study, which was to determine the effects of the NXP treatment in terms of PD.

14.) Line 239: Because no learning/memory behavioral studies were carried out, the authors cannot say that there was no toxicity associated with the aggregation. Toxicity does not necessarily imply cell loss. Aggregation within neurons in the DG/CA3 may have affected the function of the cells, which some would interpret as a toxic effect. This line needs to be changed or eliminated. The same goes for line 321 in the discussion section. This line needs to be changed or eliminated.

15.) Figure 3A: In the far right photo showing the effects of alpha synuclein, how do the authors know this is the SNpc? The authors need to show a much higher magnification. Since this reviewer is a basal ganglia morphologist, it's difficult to determine the exact location of the labeled cells. They actually appear to be dorsal to the SNpc. In addition, what cell type is labeled with p-129 in these photos?

16.) Line 250: need to change, suppresses, to 'blocks'. Also in Figure 4, there was no 4-HNE staining of the SN, it appears that the loss of TH cells following AAV-a-syn has nothing to do with oxidative stress or DNA double breaks. Please comment in the discussion section.

17.) In Figure 4 and Figure , how many hippocampal sections were taken for the cell counts? Did these sections cover the entire rostral-caudal extent of both the DG and CA3? This information should be included in the methods section. For Figure 5, where is the SN data?

18.) Discussion section: Page 16: As noted in comment #16 above, the authors need to provide an explanation as to their hypothesis as to how the dopamine cells are dying since it does not appear to be associated with either increased ROS or DNA double strand breaks. This is why a group of mice should have been removed from the study at the end of the first week following AAV-a-syn infusion, since it's possible the remaining SNpc neurons might have labeled for both markers. So how does NXP block the a-syn induced loss of DA/TH cells if not via blocking ROS or DNA double strand breaks? The authors should have carried out TH IHC in the striatum since it's highly possible that there might have been sprouting of new DA terminals in the striatum due to NXP treatment.

19.) Conclusion: the authors need to note the blocking of the aggregation was in the hippocampus and not the SNpc and that they need to tone down the idea that is drug could be used as a therapeutic since this was a neuroprotective study design.

6. PLOS authors have the option to publish the peer review history of their article (what does this mean?). If published, this will include your full peer review and any attached files.

Reviewer #1: **Yes: **Timo Myöhänen

Reviewer #2: No

---

## [Author Response · Author response to Decision Letter 0]

26 May 2022

We appreciate the constructive comments made by reviewers and giving us the opportunity for revision. We have revisited the entire manuscript and thoroughly revised the manuscript accordingly, and we are confident that our manuscript satisfies reviewers’ concerns and is greatly improved.

Prior to point-by-point answers to each question raised by reviewers, we would like to clarify the AAV-WT-α-Syn PD model that we used in this study. We noticed common concerns regarding hippocampal vs. nigral pathology and related issues. The primary aim of using this model is to test if NXP031 effectively reduces or blocks α-Syn aggregation-induced dopaminergic (DA) neuronal loss in the SN (not hippocampal dysfunction or loss). To assess the therapeutic efficacy, we measured 1) the number of TH-positive neurons in the SN, 2) pS129-α-Syn as a marker for α-Syn aggregation, and 3) 4-HNE as oxidative lipid peroxidation, and 4) γH2A.X as a DNA double-strand break. DA neuronal loss in the saline ingested group was significantly obvious compared to the NXP031 treated group. This evident DA neuronal loss invited difficulty in measuring other pathology markers (2-4) as all markers express inside cells and disappear with degeneration. Therefore, we investigate the hippocampus as an alternative region for two reasons: 1) propagation of α-Syn aggregates from the SN to the hippocampus over time is well documented (ref below), 2) hippocampal neuronal loss is minimal even though α-Syn pathology is observed (shown in our nissl staining in the hippocampus in Fig. 5). In the hippocampus, we were able to measure changes in pathologic markers between groups, confirming NXP031 blocked lipid peroxidation, α-Syn aggregation, and DNA damage. As these pathologic markers were also similarly detected in some remaining neurons in the SN (Figure below), what we observed in the hippocampus could indicate the neuroprotective mechanisms underlying how NXP031 prevents α-Syn-mediated DA neuronal degeneration.

We have attached the answer to each reviewer's question as a word file.

---

## [Decision Letter · Decision Letter 1]

21 Jun 2022

PONE-D-22-01833R1NXP031 prevents dopaminergic neuronal loss and oxidative damage in the AAV-WT-α-synuclein mouse model of Parkinson’s diseasePLOS ONE

Dear Dr. Kim,

Thank you for submitting your manuscript to PLOS ONE. After careful consideration, we feel that it has merit but does not fully meet PLOS ONE’s publication criteria as it currently stands. Therefore, we invite you to submit a revised version of the manuscript that addresses the points raised during the review process.

Reviewer has asked some clarification of figures and grammar corrections. 

We look forward to receiving your revised manuscript.

Kind regards,

Hemant K. Paudel

Academic Editor

PLOS ONE

Journal Requirements:

Reviewers' comments:

Reviewer's Responses to Questions

**Comments to the Author**

1. If the authors have adequately addressed your comments raised in a previous round of review and you feel that this manuscript is now acceptable for publication, you may indicate that here to bypass the “Comments to the Author” section, enter your conflict of interest statement in the “Confidential to Editor” section, and submit your "Accept" recommendation.

Reviewer #1: All comments have been addressed

Reviewer #2: (No Response)

2. Is the manuscript technically sound, and do the data support the conclusions?

Reviewer #1: Yes

Reviewer #2: Partly

3. Has the statistical analysis been performed appropriately and rigorously? 

Reviewer #1: Yes

Reviewer #2: Yes

4. Have the authors made all data underlying the findings in their manuscript fully available?

Reviewer #1: Yes

Reviewer #2: Yes

5. Is the manuscript presented in an intelligible fashion and written in standard English?

Reviewer #1: Yes

Reviewer #2: No

6. Review Comments to the Author

Reviewer #1: All points I have raised, have been responded by the authors. The manuscript can be accepted for publication.

Reviewer #2: June 19, 2022

Song et al have significantly revised their manuscript but this reviewer has a few more concerns that need to be addressed:

1.) Supplemental Figure 1: this reviewer found it of interest that following AAV-a-syn infusion into the SNpc, that only about 50% of the SNpc neurons were GFP labeled. I am assuming the 50% loss of TH cells after 9 weeks in the SNpc following alpha synuclein infusion were only those cells containing the alpha synuclein? Since the TH stereology was carried out throughout the entire rostral-caudal extent of the SNpc, the authors need to determine the percentage of SNpc neurons containing the alpha synuclein after just one week (since it was shown that there was essentially no loss of TH neurons at this early time period). Was the alpha synuclein induced loss of TH cells uniform throughout the entire SNpc or, since there is most likely complete loss of TH in the immediate area of the viral infusion, as essentially stated by the authors, was there significantly less loss further away from the injection site? These data are important in terms of interpreting the protective effects of the NXP031 drug administered. Since there was about a 50% loss of TH cells in this study, and if there was nearly 100% loss at the site of the infusion, this suggests that further away from the injection site, there must have been nearly no loss of TH cells. This needs to be clarified.

2.) Line 139: the authors state the animals were trained for 2 days before the start of the measurement. This reviewer assumes the mice were trained after the 9 weeks of AAV infusion? If that is the case, please state it more specifically. Initially the authors had stated the mice were trained for 3 days. Which is correct? Please clarify.

3.) Figure 2F/G: the legend states (line 244/245) the following: Representative micrographs of double staining with TH antibody and Nissl violet in the SN. (G) % of Nissl-positive neurons compared to the AAV GFP group. For this analysis, this reviewer assumes that the authors counted both TH positive and Nissl only positive cells, or were just the Nissl positive and TH NEGATIVE cells counted? In the original question asked by this reviewer, Point #11, only the Nissl positive and TH NEGATIVE cells should be counted in order to determine if there was a change in those numbers. The reason for this request is that this author has reported several times that following loss of nigrostriatal TH cells, there was an increase in the number of TH negative/Nissl positive cells in the SNpc, suggesting that some of the original TH positive cells had not died but simply were not expressing the TH protein at the moment. This reviewer has interpreted the data from Figure 2G as the authors counting all TH positive/Nissl stained cells. Please clarify.

4.) This author is still not convinced as to why the hippocampal data are included without any behavioral testing. The fact that there was no hippocampal cell loss, but there was still significant aggregation/increased p-alpha synuclein labeling, suggests to this reviewer that learning/memory deficits are most likely present. Of interest, there are recent data suggesting that during the training of a learning/memory task, there is an actual increase in DSBs in the hippocampus, suggesting that such breaks and increased repair are essential to the learning of certain tasks. Therefore, the testing of these alpha synuclein-infused mice in a L&M task is even more important. Why didn't the authors investigate p-alpha synuclein labeling in the striatum? This reviewer has found, using essentially the same AAV-alpha synuclein viral infusion into the SN, a clear increase in p-alpha synuclein in the striatum. Since the SNpc projects to the striatum, as it appears to also do to the hippocampus, it would have made more sense to investigate the striatum. Please comment.

5.) There are still some minor grammar issues that need to be corrected.

7. PLOS authors have the option to publish the peer review history of their article (what does this mean?). If published, this will include your full peer review and any attached files.

Reviewer #1: No

Reviewer #2: No

---

## [Author Response · Author response to Decision Letter 1]

27 Jun 2022

We appreciate the constructive comments made by reviewers and for giving us the opportunity for revision. We have revisited the entire manuscript and thoroughly revised the manuscript accordingly, and we are confident that our manuscript satisfies reviewers’ concerns and is greatly improved. We performed grammar corrections mentioned by the reviewer and carefully conducted the reference check journal requirements.

We have attached a file to the response to the reviewer.

---

## [Editor Report · Decision Letter 2]

13 Jul 2022

NXP031 prevents dopaminergic neuronal loss and oxidative damage in the AAV-WT-α-synuclein mouse model of Parkinson’s disease

PONE-D-22-01833R2

Dear Dr. Kim,

We’re pleased to inform you that your manuscript has been judged scientifically suitable for publication and will be formally accepted for publication once it meets all outstanding technical requirements.

Kind regards,

Hemant K. Paudel

Academic Editor

PLOS ONE
---

## [Editor Report · Acceptance letter]

19 Jul 2022

PONE-D-22-01833R2 

NXP031 prevents dopaminergic neuronal loss and oxidative damage in the AAV-WT-α-synuclein mouse model of Parkinson’s disease 

Dear Dr. Kim:

I'm pleased to inform you that your manuscript has been deemed suitable for publication in PLOS ONE. Congratulations! Your manuscript is now with our production department. 

Kind regards, 

on behalf of

Dr. Hemant K. Paudel 

Academic Editor

PLOS ONE